# Clinical Characterization of Respiratory Syncytial Virus Infection in Adults: A Neglected Disease?

**DOI:** 10.3390/v15091848

**Published:** 2023-08-31

**Authors:** Cassia F. Estofolete, Cecília A. Banho, Alice T. Verro, Flora A. Gandolfi, Bárbara F. dos Santos, Livia Sacchetto, Beatriz de C. Marques, Nikos Vasilakis, Maurício L. Nogueira

**Affiliations:** 1Laboratório de Pesquisas em Virologia, Faculdade de Medicina de São José do Rio Preto (FAMERP), São José do Rio Preto 15090-000, SP, Brazil; ceci.abanho@gmail.com (C.A.B.); florafef04@gmail.com (F.A.G.); barbara_fs13@hotmail.com (B.F.d.S.); liviasacchetto@gmail.com (L.S.); bbiacarvalhomarques@gmail.com (B.d.C.M.); 2Hospital de Base of São José do Rio Preto, São José do Rio Preto 15090-000, SP, Brazil; verroalice@gmail.com; 3Hospital da Criança e Maternidade of São José do Rio Preto, São José do Rio Preto 15091-240, SP, Brazil; 4Department of Pathology, The University of Texas Medical Branch, Galveston, TX 77555, USA; nivasila@utmb.edu; 5Department of Preventive Medicine and Population Health, The University of Texas Medical Branch, Galveston, TX 77555, USA; 6Center for Vector-Borne and Zoonotic Diseases, The University of Texas Medical Branch, Galveston, TX 77555, USA; 7Center for Biodefense and Emerging Infectious Diseases, The University of Texas Medical Branch, Galveston, TX 77555, USA; 8Center for Tropical Diseases, The University of Texas Medical Branch, Galveston, TX 77555, USA; 9Institute for Human Infection and Immunity, The University of Texas Medical Branch, Galveston, TX 77555, USA

**Keywords:** influenza A virus, influenza B virus, respiratory syncytial virus, severe acute respiratory syndrome

## Abstract

Lower respiratory tract infections (LRIs) are a significant cause of disability-adjusted life-years (DALYs) across all age groups, especially in children under 9 years of age, and adults over 75. The main causative agents are viruses, such as influenza and respiratory syncytial virus (RSV). Viral LRIs in adults have historically received less attention. This study investigated the incidence of RSV and influenza in adult patients admitted to a referral hospital, as well as the clinical profile of these infections. Molecular testing was conducted on nasopharyngeal samples taken from a respiratory surveillance cohort comprising adult (15–59 years) and elderly (60+ years) hospitalized patients who tested negative for SARS-CoV-2, to determine the prevalence for influenza and RSV. Influenza was found to be less frequent among the elderly. The main symptoms of RSV infections were cough, fever, dyspnea, malaise, and respiratory distress, while headache, nasal congestion, a sore throat, and myalgia were most frequent in influenza. Elderly patients with RSV were not found to have more severe illness than adults under age 60, underscoring the importance of providing the same care to adults with this viral infection.

## 1. Introduction

In 2019, lower respiratory tract infections (LRIs) were the fourth-leading cause of disability-adjusted life-years (DALYs) across all age groups, with the greatest impact on children aged 0–9 years, followed by adults over the age of 75. Historical data reveal similar trends, with the strongest impact on children and the elderly [1]. Furthermore, an estimated 176,740 people died from LRIs in South America in 2019 alone, and 3,872,414 life-years were lost [2].

Several viral infections cause LRIs [as reviewed in [3]], but three stand out: respiratory syncytial virus (RSV) [4], and influenza A (IAV) and B (IBV) [5], due to their case numbers, and the consequent burden on health systems due to the increased hospitalization and morbidity rates, especially among children under five years of age [4,5]. Moreover, the accurate etiological diagnosis of LRIs presents a further challenge when viral infections present overlapping symptoms [6]. Viral respiratory infections, including RSV and influenza, may manifest as various respiratory syndromes, such as bronchiolitis, wheezing, asthma exacerbation, croup, pneumonia, and pneumonitis [7].

Up until now, the research on LRIs has focused on children under five and the elderly; the host’s age is a major concern, as early severe disease develops in these groups [8]. Within this context, viral LRIs in non-elderly adults have received little attention. With the emergence of SARS-CoV-2, the unprecedented pandemic, and its devastating effects worldwide, even less attention was paid to other non-SARS-CoV-2 pathogens. Brazil had reported 484,323 cases of severe acute respiratory syndrome (SARS) by epidemiological week 47 of 2022 (ending 26 November). Of these cases, 2.2% were caused by influenza, 3.0% by other respiratory viruses, and 42.4% by undefined agents or etiologies. Among them, the age group most frequently affected was the elderly, with 216,706 cases (50.11%), with 4177 being due to influenza, and 1503 caused by other respiratory viruses [9].

To assess the circulation of respiratory viruses other than SARS-CoV-2 in adults during the COVID-19 pandemic, between December 2021 and April 2022, this study investigated the incidence of RSV and influenza in patients admitted to a referral hospital. We also evaluated the clinical characteristics of infections caused by different viral agents.

## 2. Materials and Methods

### 2.1. Description of the Cohort

This study analyzed data from a respiratory surveillance hospital cohort in São José do Rio Preto, São Paulo, Brazil, comprising enrolled adults (individuals aged 15–59 years were considered adults) and elderly patients (60+ years) who were hospitalized at the Hospital de Base de São José do Rio Preto with respiratory symptoms. Nasopharyngeal swab samples were obtained from patients presenting respiratory tract infection symptoms between December 2021 and April 2022.

### 2.2. Sample Preparation

Total RNA was extracted from 100 µL of nasopharyngeal swab samples using an Extracta Kit Fast DNA and RNA Viral Testing kit (MVXA-P096 FAST), according to the manufacturer’s instructions for the Extracta 96 DNA/RNA extractor and purifier (Loccus, Cotia, Brazil). A one-step real-time polymerase chain reaction (RT-qPCR) was performed, using primers and probes targeting the RNA-dependent RNA polymerase (RdRp), envelope (E), and nucleocapsid (N) of the SARS-CoV-2 genome and human RNAse P designed using the GeneFinder COVID-19 PLUS RealAmp Kit. (OSANG Healthcare, Anyang, Republic of Korea).

Samples that yielded negative results for COVID-19 were tested for three different respiratory viruses (IAV, IBV, and RSV), using an Allplex SARS-CoV-2/IAV/IBV/RSV assay (Seegene Inc., Seoul, Republic of Korea), according to the manufacturer’s instructions. All samples that tested positive for COVID-19 were excluded from the study. RT-qPCR was conducted with a QuantStudio 3 Real-Time PCR System (Thermo Fisher Scientific, Waltham, MA, USA). The results were interpreted based on the cycle threshold (Ct), with samples presenting Cq ≤ 40 considered positive. Positive and negative controls included in the GeneFinder COVID-19 Plus RealAmp kit and the Allplex SARS-CoV-2/IAV/IBV/RSV assay were used, in addition to positive samples included as a positive control. All samples that tested negative for SARS-CoV-2, but positive for other viruses, were included (Figure 1).

Demographic and clinical data, such as sex, ethnicity, age, education level, comorbidities, signs/symptoms, and severity were obtained from the participants’ electronic medical records or local reporting forms and subjected to descriptive and inferential analysis. The chi-square test or Fisher’s exact test was used to compare the categorical variables for the groups diagnosed with IAV, IBV, and RSV, with *p* < 0.05 considered statistically significant. All statical analyses were performed using the SPSS Statistics software package (version 28.0.1.1; IBM Corporation, Armonk, NY, USA).

## 3. Results

During the specified study period, 110 samples were considered eligible for the study (Figure 1 and Table 1): 36 (32.73%) were positive for the IAV virus, and 9 (8.18%) for IBV, and RSV was most prevalent, with 51 (46.36%; *p* < 0.001). Although they were not included in the final analysis of this study, coinfection with the three viruses of interest was observed in 14 samples (12.73%).

The patients infected with IAV were predominantly adults (83.33%; *p* < 0.001), and white (87.88%; *p* < 0.001), with a mean age of 38.56 years (±19.51). No significant statistical differences were observed with respect to sex, education, or the presence of comorbidities. For patients infected with IBV, no significant differences were noted, although they were predominantly female (77.78%), white (100%) adults (77.78%), with a mean age of 42.89 years (±18.19), and education up through high school (57.14%). No age group or sex was predominant in the RSV patient group, but white ethnicity (90.20%; *p* < 0.001), education up to elementary school (56.52%; *p* = 0.004), and the presence of comorbidities (84.78%; *p* < 0.001) were more prevalent; the mean age was 59.20 years (±18.21) (Appendix A).

The clinical data were subdivided according to the diagnosis of influenza A/B (Flu) or RSV, as shown in Table 2 and complemented by Appendix A. Influenza appeared less frequently in the elderly participants. We found a significant difference in the mean ages: RSV-infected patients were older (59.2 years vs. 39.42 years for patients with influenza). No significant differences were found for sex or ethnicity.

Comorbidities were more frequently present in RSV-infected patients (79.59%) than in other groups (*p* < 0.001). Of these, high blood pressure was most common in both groups (flu = 18.42%; RSV = 38.78%), but less frequent in the patients with influenza (*p* = 0.024). An additional statistically significant comorbidity was cardiopathy, which was observed more frequently in patients with RSV (*p* = 0.004). The frequency of comorbidities by infection group is shown in Table 2.

The main symptoms of influenza were (in descending order) coughing (78.38%), fever (59.46%), nasal congestion (56.76%), myalgia (51.35%), and headache (45.95%). In RSV infections, coughing (71.43%), fever (38.1%), dyspnea (38.1%), malaise (35.71%), and respiratory distress (35.71%) were predominant. Group analysis indicated that headache (*p* = 0.021), nasal congestion (*p* = 0.003), sore throat (*p* > 0.001), and myalgia (*p* = 0.003) were more frequent in the flu group, while potentially severe symptoms, such as respiratory distress (*p* < 0.001) and dyspnea (*p* = 0.005), were more frequent in the RSV group, as shown in Table 2. In terms of severity, no significant difference was seen in deaths or the need for intensive care, but ventilatory support was more frequently required in patients with RSV (*p* < 0.001). These variables also appear in Table 2.

Our subsequent analysis only examined the hospitalized adults; the most frequently identified respiratory agent in this group was IAV (42.25%), followed by RSV (32.39%), coinfections (15.49%), and IBV (9.86%) (*p* < 0.001) (Table 3). At this stage, the coinfection group was excluded, as it was impossible to define which agent influenced the variables. No differences were observed with regard to sex or ethnicity, while adults with an incomplete primary school education were predominant in the flu group (*p* = 0.037). Comorbidities were generally more frequent in the RSV group (*p* = 0.042) (Table 3).

When symptoms were associated with the viral agent, fever (62.50%; *p* = 0.012), headache (53.13%; *p* = 0.008), and myalgia (56.25%; *p* = 0.005) were seen to be more frequent in the flu group, while dyspnea (42.11%; *p* = 0.003) and respiratory distress (31.58%; *p* = 0.002) were, again, associated with RSV infection. Ventilatory support was more frequently required in RSV-infected patients (26.09%; *p* = 0.009), consistent with the higher frequency of dyspnea and respiratory distress (Table 4).

A final analysis compared the clinical manifestations of RSV infection between adults and elderly patients (Table 5). No differences were observed in sex or ethnicity, while elderly patients with an incomplete primary school education were most prevalent (69.23%; *p* = 0.049), as was the presence of comorbidities (*p* = 0.001), most notably cardiopathy (*p* = 0.046) and high blood pressure (*p* = 0.049). The most common symptoms in adults (in descending order) were coughing (89.47%), dyspnea (42.11%), respiratory distress (31.78%), fever (26.32%), and nasal congestion (26.32%). The elderly patients were more likely to have a cough (56.42%), fever (47.83%), malaise (47.83%), respiratory distress (39.13%), and dyspnea (34.78%). Differences were only observed for coughing (*p* = 0.037) and diarrhea (*p* = 0.035), which were more common in the adults. Interestingly, no differences were observed for dyspnea, respiratory distress, ventilatory support, or death.

## 4. Discussion

This study reports the frequency of IAV, IBV, and RSV as the cause of LRIs in patients hospitalized between December 2021 and April 2022. This is not the typical season for such respiratory infections in the Southern Hemisphere, and the COVID-19 pandemic was still underway in Brazil. We found RSV to be the main causative agent among the respiratory viruses for which we tested, associated with dyspnea, respiratory distress, and the need for ventilator support, while influenza caused disease characterized by milder symptoms. Most notably, no significant differences were reported in how RSV infections manifested among adult and elderly patients.

RSV is recognized as an important cause of respiratory infection, particularly in children and the elderly. In children, RSV may manifest as an upper respiratory tract infection, characterized by nasal congestion, coughing, fever, malaise, poor appetite, and dehydration. About one-third of infants may develop LRIs that cause pneumonia, bronchiolitis, and laryngotracheitis [10]. These patients may present with tachypnea, wheezing, noisy breathing, and even apnea, progressing to respiratory failure and death [10,11]. The presence of fever is not imperative to identifying suspected cases in RSV, nor in influenza, as proposed by the WHO, as it is absent in around 50% of cases in children and elderly patients [12]. RSV is also associated with long-term sequelae, such as asthma, recurrent wheezing, atopy, and allergies, as well as abnormal respiratory function due to airway remodeling [10]. A review of the costs associated with pediatric hospitalizations in the United States from 2014 to 2021 determined that the average cost per RSV hospitalization ranged from USD 10,214 to USD 57,406, depending on the age group [13].

The literature highlights the involvement of RSVs in children, as well as in elderly people and adults with comorbidities, particularly immunosuppression, asthma, chronic obstructive pulmonary disease, and congestive heart failure [14], in whom this virus may represent an important concern. In these individuals, RSV infection may manifest as nasal congestion and coughing, wheezing, ear pain, sinusitis, crackles, infiltrates on chest radiographs, pneumonia, and respiratory distress. As in children, longer-term complications may be observed, such as a permanent decline in the respiratory function [10]. Falsey et al. observed an annual incidence of RSV ranging from 4 to 10% in high-risk adults, with a higher utilization of medical care; in contrast, the annual frequency in the elderly population was 3–7% [15]. A study in a Thai hospital cohort of adults aged 15 years and above reported 69 cases of RSV infection [16]. These patients were mostly above the age of 50 (87%), and all had at least one comorbidity. The most common clinical presentation was community-acquired pneumonia (82.6%), followed by asthma or chronic obstructive pulmonary disease exacerbation, and acute bronchitis. An analysis of the costs of RSV-associated hospitalization in the elderly estimated an average between USD 8241 and USD 16,034 [17,18]. In adults, the average cost was USD 11,124 [18].

In our study, RSV was identified in over 40% of the samples from non-COVID-19-SARS cases, with similar frequencies in the adult and elderly populations (45.1% vs. 54.9%). Dyspnea was observed in around 38%, and respiratory distress in around 35%, highlighting the clinical impacts of the disease. Comorbidities in general were more frequent in the elderly, as expected. Immunosuppression and pulmonary or cardiac disease are also more common in the elderly, and could represent a risk factor for severe RSV infection. However, the analysis of the clinical manifestation and outcomes of RSV infection did not reveal differences in the occurrence of dyspnea, respiratory distress, or the need for ventilatory support between the adult and elderly groups. Even when comorbidities that could potentially raise the risk for severe disease were present, RSV infection manifested with the same potential severity in both adult and elderly patients, causing dyspnea, respiratory distress, and requiring ventilatory support.

Meanwhile, influenza accounted for 40.91% of cases in this study. Adults were more affected than the elderly, especially by IAV. It is important to note that Brazil’s national immunization program recommends flu vaccination for at-risk groups, including the elderly and adults with comorbidities, but not for healthy adults [19]. The mean age of the patients with IAV and IBV was also lower than that of the RSV patients, and the frequency of comorbidities was consistent with age. The clinical symptom data showed that influenza produced milder disease than RSV, with headaches, myalgia, nasal congestion, and sore throat seen most frequently; these symptoms are usually associated with the upper respiratory tract. In contrast, RSV was associated with a higher frequency of dyspnea and respiratory distress. Our data reinforce that RSV represents a higher potential severity than influenza, a finding corroborated by the more frequent need for ventilatory support.

Influenza was the most common infection when only adult patients were analyzed. The infection was also milder compared to RSV, with fever, headache, myalgia, and nasal congestion most frequently manifesting. In general, IAV and IBV infections may be mostly asymptomatic, with some upper respiratory tract symptoms, such as fever, chills, myalgia, malaise, a dry cough, a sore throat, and nasal discharge [20,21]. The US Centers for Disease Control and Prevention (CDC) estimates the rates of hospitalization and death in symptomatic infections at 1% and 0.1%, respectively [22]. Severe influenza may include respiratory complications such as bronchiolitis, bronchitis, pneumonia, bacterial coinfections, respiratory failure, and acute respiratory distress syndrome, or non-respiratory developments, including heart involvement, myositis, aseptic meningitis, encephalomyelitis, and Guillain–Barré syndrome [20,21]. A four-year prospective hospital cohort involving RSV and influenza revealed similar mortality rates in both diseases: 8% and 7%, respectively [15]. Although we found higher rates, we did not find a statistically significant difference between these viruses. Even though influenza may have the potential for severe disease, we found a higher frequency of this outcome in cases of RSV infection (*p* < 0.001 for respiratory distress, dyspnea, and ventilatory support). Even in adults, 6.25% of infected patients reported dyspnea, 5.56% required ventilatory support, and 2.78% needed intensive care.

Seasonal outbreaks of influenza and RSV tend to occur in both tropical and temperate countries in winter [23,24], but summertime cases, such as the ones seen in our cohort, do not represent an unexplained event, as even influenza and RSV are common in that season. Non-pharmaceutical interventions to control the spread of SARS-CoV-2, such as masking, also effectively limited the transmission of other respiratory agents, such as RSV and influenza. Cases of viral respiratory illnesses rebounded as control measures loosened, even out of season; this effect was also observed in other countries, including Australia, New Zealand, and South Africa [25]. Similarly, a flu outbreak occurred in Rio de Janeiro in November 2021, when lower mean temperatures were observed, alongside a discrepancy between flu vaccine strains and the circulating virus [23].

## 5. Conclusions

Our findings demonstrate the importance of differential diagnostics for respiratory infections, even during outbreaks and epidemics. Although the literature has adequately described the effects of respiratory agents in children and the elderly, in our cohort, we did not observe different clinical manifestations of RSV infections in adult and elderly patients, suggesting that this disease may have a significant clinical impact in adults in general, and not just in known risk groups. This observation underscores the need to define the agents that cause moderate and severe respiratory infections, especially as preventive and therapeutic options emerge. Previous studies have been supporting clinical trials in newborns and elderly (for example, clinicaltrials.gov: NCT04908683; NCT04908683; NCT05559476; NCT04732871), but they have not been focusing on adults, although these comprise a potential severity group that should be targeted through future vaccination programs. This is highlighted in this study through our demonstration of how dyspnea, respiratory distress, and the need for ventilatory support occurred at similar frequencies in adults and the elderly.

## Figures and Tables

**Figure 1 viruses-15-01848-f001:**
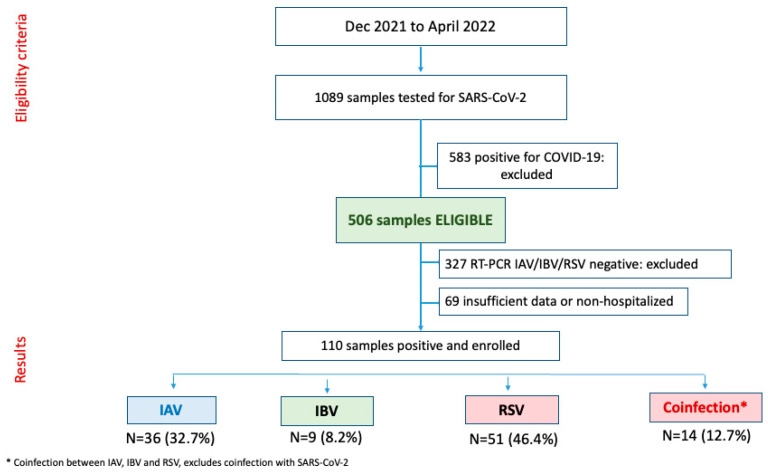
Selection flow of samples and distribution of respiratory agents in the hospital cohort.

**Table 1 viruses-15-01848-t001:** Epidemiological data from 110 patients with samples showing infection with influenza A, influenza B, and respiratory syncytial virus in the respiratory surveillance cohort.

	Total Samples (N = 110)					
	N Response	N Positive or Mean	% or s.d.	*p*-Value	*X* ^2^	df
Agent						
RSV	110	51	46.36%	<0.001	41.782	3
Influenza A	110	36	32.73%			
Influenza B	110	9	8.18%			
Coinfection	110	14	12.73%			
Sex						
Female	110	52	47.27%	0.567	0.327	1
Male	110	58	52.73%			
Age group						
Mean age	110	50.38	20.42	-	-	-
Adult	110	71	64.55%	0.002	9.309	1
Elderly	110	39	35.45%			
Ethnicity						
White	104	94	90.38%	<0.001	152.558	2
Black	104	7	6.73%			
Mixed race	104	3	2.88%			
Education						
Primary incomplete	93	40	43.01%	0.115	4.323	2
Primary	93	0	0.00%			
High school	93	24	25.81%			
College	93	29	31.18%			
Comorbidities	97	62	63.92%	0.006	7.515	1
Cardiopathy	97	16	16.49%	<0.001	43.557	1
Blood pressure	97	34	35.05%	0.003	8.670	1
Hematological disease	97	3	3.09%	<0.001	85.371	1
Down syndrome	97	2	2.06%	<0.001	89.165	1
Liver disease	97	6	6.19%	<0.001	74.485	1
Asthma	97	1	1.03%	<0.001	93.041	1
Diabetes	97	20	20.62%	<0.001	33.495	1
Neurovascular disease	97	7	7.22%	<0.001	71.021	1
Neurological disease	97	5	5.15%	<0.001	78.031	1
Pneumopathy	97	7	7.22%	<0.001	71.021	1
Immunosuppression	97	1	1.03%	<0.001	93.041	1
Kidney disease	97	7	7.22%	<0.001	71.021	1
Obesity	97	5	5.15%	<0.001	78.031	1
Smoker	85	16	18.82%	<0.001	33.047	1
Cancer	97	9	9.28%	<0.001	64.340	1

df: degrees of freedom.

**Table 2 viruses-15-01848-t002:** Clinical characteristics of 96 patients infected with influenza and respiratory syncytial virus in the respiratory surveillance cohort.

	Flu			RSV			
	N Response	N Positive or Mean	% or s.d.	N Response	N Positive or Mean	% or s.d.	*p*-Value
Age group							
Adult (15–59 years)	45	37	82.22%	51	23	45.10%	<0.001
Elderly (60+ years)	45	8	17.78%	51	28	54.90%	
Mean age (years)	45	39.42	19.13	51	59.20	18.21	<0.001
Sex							
Female	45	21	46.67%	51	26	50.98%	0.673
Male	45	24	53.33%	51	25	49.02%	
Ethnicity							
White	41	37	90.24%	51	46	90.20%	0.894
Black	41	3	7.32%	51	3	5.88%	
Mixed race	41	1	2.44%	51	2	3.92%	
Education							
Primary incomplete	38	7	18.42%	46	26	56.52%	0.002
Primary	38	0	0.00%	46	0	0.00%	
High school	38	15	39.47%	46	9	19.57%	
College	38	16	42.11%	46	11	23.91%	
Comorbidities	38	18	47.37%	49	39	79.59%	<0.001
Cardiopathy	38	1	2.63%	49	14	28.57%	<0.001
Blood pressure	38	7	18.42%	49	19	38.78%	0.024
Hematological disease	38	0	0.00%	49	2	4.08%	0.499
Down syndrome	38	0	0.00%	49	2	4.08%	0.499
Liver disease	38	0	0.00%	49	6	12.24%	0.300
Asthma	38	1	2.63%	49	0	0.00%	0.452
Diabetes	38	4	10.53%	49	13	26.53%	0.044
Neurovascular disease	38	0	0.00%	49	6	12.24%	0.030
Neurological disease	38	0	0.00%	49	5	10.20%	0.061
Pneumopathy	38	1	2.63%	49	5	10.20%	0.215
Immunosuppression	38	0	0.00%	48	1	2.08%	1
Kidney disease	38	0	0.00%	49	6	12.24%	0.030
Obesity	38	1	2.63%	49	3	6.12%	0.623
Smoker	35	6	17.14%	41	7	17.07%	1
Cancer	38	4	10.53%	49	5	10.20%	1
Signs and symptoms							
Fever	37	22	59.46%	42	16	38.10%	0.058
Headache	37	17	45.95%	42	9	21.43%	0.021
Myalgia	37	19	51.35%	42	8	19.05%	0.003
Nasal congestion	37	21	56.76%	42	10	23.81%	0.003
Cough	37	29	78.38%	42	30	71.43%	0.478
Sore throat	37	15	40.54%	42	4	9.52%	<0.001
Dyspnea	37	4	10.81%	42	16	38.10%	0.005
Respiratory distress	37	1	2.70%	42	15	35.71%	<0.001
Diarrhea	37	2	5.41%	42	4	9.52%	0.679
Vomiting	37	0	0.00%	42	1	2.38%	1
Abdominal pain	37	3	8.11%	42	2	4.76%	0.661
Malaise	37	8	21.62%	42	15	35.71%	0.169
Loss of smell	37	1	2.70%	42	0	0.00%	0.468
Loss of taste	37	1	2.70%	42	0	0.00%	0.468
Hospitalization	45	45	100.00%	51	51	100.00%	NA
ICU	42	1	2.38%	51	5	9.80%	0.217
Ventilatory support	42	4	9.52%	50	23	46.00%	<0.001
Death	43	8	18.60%	46	15	32.61%	0.132

**Table 3 viruses-15-01848-t003:** Epidemiological data from 71 adult patients infected with influenza A, influenza B, and respiratory syncytial virus in the respiratory surveillance cohort.

	Samples from Hospitalized Adults					
	N Response	N Positive or Mean	% or s.d.	*p*-Value	*X* ^2^	df
Agent						
RSV	71	23	32.39%	<0.001	19	3
Influenza A	71	30	42.25%			
Influenza B	71	7	9.86%			
Coinfection	71	11	15.49%			
Sex						
Female	71	31	43.66%	0.285	1.141	1
Male	71	40	56.34%			
Age group						
Mean age	71	38	13.20	-	-	-
Ethnicity						
White	67	58	86.57%	<0.001	85.642	2
Black	67	6	8.96%			
Mixed race	67	3	4.48%			
Education						
Primary incomplete	63	16	25.40%	0.692	0.737	2
Primary	63	0	0.00%			
High school	63	20	31.75%			
College	63	21	33.33%			
Comorbidities	63	28	44.44%	<0.001	0.778	1
Cardiopathy	63	3	4.76%	<0.001	51.571	1
Blood pressure	63	15	23.81%	<0.001	17.286	1
Hematological disease	63	2	3.17%	<0.001	55.254	1
Down syndrome	63	2	3.17%	<0.001	55.524	1
Liver disease	63	4	6.35%	<0.001	48.016	1
Asthma	63	0	0.00%			
Diabetes	63	9	14.29%	<0.001	32.143	1
Neurovascular disease	63	1	1.59%	<0.001	59.063	1
Neurological disease	63	0	0.00%			
Pneumopathy	63	3	4.76%	<0.001	51.571	1
Immunosuppression	63	1	1.59%	<0.001	59.063	1
Kidney disease	63	4	6.35%	<0.001	48.016	1
Obesity	63	3	4.76%	<0.001	51.571	1
Smoker	53	10	18.87%	<0.001	20.547	1
Cancer	63	4	6.35%	<0.001	48.016	1
Symptoms						
Fever	60	31	51.67%	0.796	0.067	1
Headache	60	25	41.67%	0.197	1.667	1
Myalgia	60	36	60.00%	0.302	1.067	1
Nasal congestion	60	30	50.00%	1	0	1
Cough	60	48	80.00%	<0.001	21.6	1
Sore throat	60	20	33.33%	0.010	6.667	1
Dyspnea	60	12	20.00%	<0.001	21.6	1
Respiratory distress	60	7	11.67%	<0.001	35.267	1
Diarrhea	60	5	8.33%	<0.001	41.667	1
Vomiting	60	0	0.00%	NA	NA	NA
Abdominal pain	60	4	6.67%	<0.001	45.067	1
Malaise	60	11	18.33%	<0.001	24.067	1
Loss of smell	60	2	3.33%	<0.001	52.267	1
Loss of taste	60	2	3.33%	<0.001	52.267	1
Hospitalization	71	71	100.00%	NA	NA	NA
ICU	70	5	7.14%	<0.001	51.429	1
Ventilatory support	69	12	17.39%	<0.001	29.348	1
Death	57	3	5.26%	<0.001	18.284	1

**Table 4 viruses-15-01848-t004:** Clinical and epidemiological data from 60 adult patients infected with influenza and respiratory syncytial virus in the respiratory surveillance cohort.

	Flu (A and B)			RSV			
	N Response	N Positive or Mean	% or s.d.	N Response	N Positive or Mean	% or s.d.	*p*-Value
Mean age (years)	37	31.97	10.72	23	43.26	13.29	0.001
Sex							
Female	37	17	45.95%	23	11	47.83%	0.887
Male	37	20	54.05%	23	12	52.17%	
Ethnicity							
White	34	30	88.24%	23	19	82.61%	0.633
Black	34	3	8.82%	23	2	8.70%	
Mixed race	34	1	2.94%	23	2	8.70%	
Education							
Primary incomplete	30	3	10.00%	20	8	40.00%	0.037
Primary	30	0	0.00%	20	0	0.00%	
High school	30	13	43.33%	20	7	35.00%	
College	30	14	46.67%	20	5	25.00%	
Comorbidities	33	10	30.30%	20	13	65.00%	0.013
Cardiopathy	33	0	0.00%	20	3	15.00%	0.049
Blood pressure	33	5	15.15%	20	5	25.00%	0.475
Hematological disease	33	0	0.00%	20	1	5.00%	0.377
Down syndrome	33	0	0.00%	20	2	10.00%	0.138
Liver disease	33	0	0.00%	20	4	20.00%	0.017
Asthma	33	0	0.00%	20	0	0.00%	NA
Diabetes	33	1	3.03%	20	5	25.00%	0.024
Neurovascular disease	33	0	0.00%	20	1	5.00%	0.377
Neurological disease	33	0	0.00%	20	0	0.00%	NA
Pneumopathy	33	0	0.00%	20	2	10.00%	0.138
Immunosuppression	33	0	0.00%	20	1	5.00%	0.377
Kidney disease	33	0	0.00%	20	3	15.00%	0.049
Obesity	33	1	3.03%	20	1	5.00%	1
Smoker	28	4	14.29%	17	3	17.65%	1
Cancer	33	3	9.09%	20	1	5.00%	1
Symptoms							
Fever	32	20	62.50%	19	5	26,32%	0.012
Headache	32	17	53.13%	19	3	15.79%	0.008
Myalgia	32	18	56.25%	19	3	15.79%	0.005
Nasal congestion	32	19	59.38%	19	5	26.32%	0.022
Cough	32	24	75.00%	19	17	89.47%	0.287
Sore throat	32	13	40.63%	19	3	15.79%	0.065
Dyspnea	32	2	6.25%	19	8	42.11%	0.003
Respiratory distress	32	0	0.00%	19	6	31.58%	0.002
Diarrhea	32	1	3.13%	19	4	21.05%	0.058
Vomiting	32	0	0.00%	19	0	0.00%	NA
Abdominal pain	32	3	9.38%	19	1	5.26%	1
Malaise	32	6	18.75%	19	4	21.05%	1
Loss of smell	32	1	3.13%	19	0	0.00%	1
Loss of taste	32	1	3.13%	19	0	0.00%	1
Hospitalization	37	37	100.00%	23	23	100.00%	NA
ICU	36	1	2.78%	23	3	13.04%	0.289
Ventilatory support	36	2	5.56%	23	6	26.09%	0.009
Death	36	6	16.67%	20	8	40.00%	0.053

**Table 5 viruses-15-01848-t005:** Clinical manifestations of RSV in hospitalized adult and elderly patients.

	Adult			Elderly			
	N Response	N Positive or Mean	% or s.d.	N Response	N Positive or Mean	% or s.d.	*p*-Value
Mean age (years)	23	43.26	13.29	28	72.29	8.74	<0.001
Sex							
Female	23	11	47.83%	28	15	53.57%	0.683
Male	23	12	52.17%	28	13	46.43%	
Ethnicity							
White	23	19	82.61%	28	27	96.43%	0.195
Black	23	2	8.70%	28	1	3.57%	
Mixed race	23	2	8.70%	28	0	0.00%	
Education							
Primary incomplete	20	8	40.00%	26	18	69.23%	0.049
Primary	20	0	0.00%	26	0	0.00%	
High school	20	7	35.00%	26	2	7.69%	
College	20	5	25.00%	26	6	23.08%	
Comorbidities	20	13	65.00%	26	26	100.00%	0.001
Cardiopathy	20	3	15.00%	26	11	42.31%	0.046
Blood pressure	20	5	25.00%	26	14	53.85%	0.049
Hematological disease	20	1	5.00%	26	1	3.85%	1
Down syndrome	20	2	10.00%	26	0	0.00%	0.184
Liver disease	20	4	20.00%	26	2	7.69%	0.380
Asthma	20	0	0.00%	26	0	0.00%	NA
Diabetes	20	5	25.00%	26	8	30.77%	0.667
Neurovascular disease	20	1	5.00%	26	5	19.23%	0.212
Neurological disease	20	0	0.00%	26	5	19.23%	0.059
Pneumopathy	20	2	10.00%	26	3	11.54%	1
Immunosuppression	20	1	5.00%	26	0	0.00%	0.435
Kidney disease	20	3	15.00%	26	3	11.54%	1
Obesity	20	1	5.00%	26	2	7.69%	1
Smoker	17	3	17.65%	26	4	15.38%	1
Cancer	20	1	5.00%	26	4	15.38%	0.369
Signs and Symptoms							
Fever	19	5	26.32%	23	11	47.83%	0.153
Headache	19	3	15.79%	23	6	26.09%	0.477
Myalgia	19	3	15.79%	23	5	21.74%	0.709
Nasal congestion	19	5	26.32%	23	5	21.74%	1
Cough	19	17	89.47%	23	13	56.52%	0.037
Sore throat	19	3	15.79%	23	1	4.35%	0.313
Dyspnea	19	8	42.11%	23	8	34.78%	0.627
Respiratory distress	19	6	31.58%	23	9	39.13%	0.611
Diarrhea	19	4	21.05%	23	0	0.00%	0.035
Vomiting	19	0	0.00%	23	1	4.35%	1
Abdominal pain	19	1	5.26%	23	1	4.35%	1
Malaise	19	4	21.05%	23	11	47.83%	0.071
Loss of smell	19	0	0.00%	23	0	0.00%	NA
Loss of taste	19	0	0.00%	23	0	0.00%	NA
Hospitalization	23	23	100.00%	28	28	100.00%	NA
ICU	23	3	13.04%	28	2	7.14%	0.647
Ventilatory support	23	6	26.09%	27	15	55.56%	0.142
Death	20	8	40.00%	26	7	26.92%	0.348

## Data Availability

De-identified data are available from the authors upon request.

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
