# Peer review of "Clinical Characterization of Respiratory Syncytial Virus Infection in Adults: A Neglected Disease?"

_viruses, 2023, doi:10.3390/v15091848_

Round 1

Reviewer 1 Report

The method used and described in the materials and methods is not suitable. Lines 68 to 71 describe the use of QuantStudio 3 as an instrument while the Allplex SARS-CoV-2/IAV/IBV/RSV assay amplification kit is validated on CFX96. In order to be used for epidemiological purposes, the kits must be used according to certification.

Author Response

Attached letter to reviewers. Thanks for considerations. 

Reviewer 2 Report

Introduction: recommended to improve the background providing some info regarding Flu and RSV epidemiology in elderly.

Discussion: not clear message about relationship between RSV infection and severity of clinical features and age. 

Limited sample size. IBV infected very poorly represented.

It is not so clear if the period of enrollment is expected to be associated to RSV outbreak or not, while it is clear for Flu: I would suggest to clarify.

Is the final recommendation to include adults in the trial for RSV vaccination? I would expand little bit the indications.

Author Response

(The authors gave the same response as above.)

Reviewer 3 Report

Estofolete et al., described comparison of clinical features among Flu and RSV infections in adult and elderly people. The analysis of RSV infection in adult (including elderly people) is valuable and the results are interesting. I have several comments to improve manuscript.

1.                   Overall, it is difficult to understand the tables. Are the presentations able to be devised to make it easier to understand? What are the statistical parameters for comorbidities meant in Table 1? What is the meaning of Table 3 in the story of this article?

2.                   It seems Allplex kits are adjusted to BioRad CFX96DX. Is there any validation for adjusting it to QS3? Who validated the cut off value of this kit in QS3? I think the default parameter in the analysis with QS3 is Ct. Why the author uses Cq? The 2nd derivative max method is not supported in Thermo instruments, so it seems it is not required to consider mixed use of Ct and Cp.

3.                   Why did the author not perform the comparison of elderly people between FluA and RSV? Certainly, the number of FluA in elderly is small, but it seems very important.

4.                   It seems the trend of difference in clinical symptoms between Flu and RSV even in adult. In global surveillance of RSV planned by WHO, the criteria SARI without fever and ILI without fever are applied not to miss RSV cases. Therefore, it is important to show that the fever is also low in RSV infection even in adult and elderly people.

5.                   Recent approved RSV vaccines (GSK and Pfizer) are for elderly people (60+ years). It is important to mention the necessity of vaccines in adult people.

Minor:

The ventilatory support of RSV (0.009) in no bold in table 4.

Author Response

(The authors gave the same response as above.)

Reviewer 4 Report

The paper describes LRI in adults and the elderly. Yet the sample acquired is more so related to URI in patients.  It can be used for some LRI, but references to previous research comparing and contrasting the sample used and its appropriateness are not addressed for the adult population. BAL fluid and sputum are more associated with LRI sample collection.

How can the researcher be confident that this is a LRI and not a URI?

Author Response

(The authors gave the same response as above.)

Round 2

Reviewer 3 Report

I agree to the modification.